# Understanding Neural Network Binarization with Forward and Backward Proximal Quantizers

**Yiwei Lu**[*]
School of Computer Science
University of Waterloo
Vector Institute
yiwei.lu@uwaterloo.ca

**Yaoliang Yu**
School of Computer Science
University of Waterloo
Vector Institute
yaoliang.yu@uwaterloo.ca

**Xinlin Li**
Huawei Noah's Ark Lab
xinlin.li1@huawei.com

**Vahid Partovi Nia**
Huawei Noah's Ark Lab
vahid.partovinia@huawei.com

## Abstract

In neural network binarization, BinaryConnect (BC) and its variants are considered the standard. These methods apply the sign function in their forward pass and their respective gradients are backpropagated to update the weights. However, the derivative of the sign function is zero whenever defined, which consequently freezes training. Therefore, implementations of BC (e.g., BNN) usually replace the derivative of sign in the backward computation with identity or other *approximate gradient* alternatives. Although such practice works well empirically, it is largely a heuristic or "training trick." We aim at shedding some light on these training tricks from the optimization perspective. Building from existing theory on ProxConnect (PC, a generalization of BC), we (1) equip PC with *different* forward-backward quantizers and obtain ProxConnect++ (PC++) that includes existing binarization techniques as special cases; (2) derive a principled way to synthesize forward-backward quantizers with automatic theoretical guarantees; (3) illustrate our theory by proposing an enhanced binarization algorithm BNN++; (4) conduct image classification experiments on CNNs and vision transformers, and empirically verify that BNN++ generally achieves competitive results on binarizing these models.

## 1 Introduction

The recent success of numerous applications in machine learning is largely fueled by training big models with billions of parameters, e.g., GPTs in large language models [7, 8], on extremely large datasets. However, as such models continue to scale up, end-to-end training or even fine-tuning becomes prohibitively expensive, due to the heavy amount of computation, memory and storage required. Moreover, even after successful training, deploying these models on resource-limited devices or environments that require real-time inference still poses significant challenges.

A common way to tackle the above problems is through model compression, such as pruning [44, 47, 51], reusing attention [6], weight sharing [57], structured factorization [49], and network quantization [16, 30, 32, 38]. Among them, network quantization (i.e., replacing full-precision weights with lower-precision ones) is a popular approach. In this work we focus on an extreme case of network quantization: binarization, i.e., constraining a subset of the weights to be only binary (i.e., $\pm 1$), with

---

[*]Work done during an internship at Huawei Noah's Ark Lab.

37th Conference on Neural Information Processing Systems (NeurIPS 2023).

the benefit of much reduced memory and storage cost, as well as inference time through simpler and faster matrix-vector multiplications, which is one of the main computationally expensive steps in transformers and the recently advanced vision transformers [14, 34, 52].

For neural network binarization, BinaryConnect [BC, 11] is considered the de facto standard. BC applies the sign function to binarize the weights in the forward pass, and evaluates the gradient at the binarized weights using the Straight Through Estimator [STE, 4][2]. This widely adopted training trick has been formally justified from an optimization perspective: Dockhorn et al. [13], among others, identify BC as a nonconvex counterpart of dual averaging, which itself is a special case of the generalized conditional gradient algorithm. Dockhorn et al. [13] further propose ProxConnect (PC) as an extension of BC, by allowing arbitrary proximal quantizers (with sign being a special case) in the forward pass.

However, practical implementations [e.g., 2, 12, 22] usually apply an *approximate gradient* of the sign function on top of STE. For example, Hubara et al. [22] employ the hard tanh function as an approximator of sign. Thus, in the backward pass, the derivative of sign is approximated by the indicator function $\mathbf{1}_{[-1,1]}$, the derivative of hard tanh. Later, Darabi et al. [12] consider the sign-Swish function as a more accurate and flexible approximation in the backward pass (but still employs the sign in the forward pass).

Despite their excellent performance in practice, *approximate gradient* approaches cannot be readily understood in the PC framework of Dockhorn et al. [13], which does not equip any quantization in the backward pass. Thus, the main goal of this work is to further generalize PC and improve our understanding of approximate gradient approaches. Specifically, we introduce PC++ that comes with a *pair* of forward-backward proximal quantizers, and we show that most of the existing approximate gradient approaches are special cases of our proximal quantizers, and hence offering a formal justification of their empirical success from an optimization perspective. Moreover, inspired by our theoretical findings, we propose a novel binarization algorithm BNN++ that improves BNN+ [12] on both theoretical convergence properties and empirical performances. Notably, our work provides direct guidance on designing new forward-backward proximal quantizers in the PC++ family, with immediate theoretical guarantees while enabling streamlined implementation and comparison of a wide family of existing quantization algorithms.

Empirically, we benchmark existing PC++ algorithms (including the new BNN++) on image classification tasks on CNNs and vision transformers. Specifically, we perform weight (and activation) binarization on various datasets and models. Moreover, we explore the fully binarized scenario, where the dot-product accumulators are also quantized to 8-bit integers. In general, we observe that BNN++ is very competitive against existing approaches on most tasks, and achieves 30x reduction in memory and storage with a modest 5-10% accuracy drop compared to full precision training.

We summarize our main contributions in more detail:

- We generalize ProxConnect with forward-backward quantizers and introduce ProxConnect++ (PC++) that includes existing binarization techniques as special cases.

- We derive a principled way to synthesize forward-backward quantizers with theoretical guarantees. Moreover, we design a new BNN++ variant to illustrate our theoretical findings.

- We empirically compare different choices of forward-backward quantizers on image classification benchmarks, and confirm that BNN++ is competitive against existing alternatives.

## 2 Background

In neural network quantization, we aim at minimizing the usual (nonconvex) objective function $\ell(\mathbf{w})$ with discrete weights $\mathbf{w}$:

$$\min_{\mathbf{w} \in Q} \ell(\mathbf{w}), \tag{1}$$

where $Q \subseteq \mathbb{R}^d$ is a discrete, nonconvex quantization set such as $Q = \{\pm 1\}^d$. The acquired discrete weights $\mathbf{w} \in Q$ are compared directly with continuous full precision weights, which we denote

---

[2]Note that we refer to STE as its original definition by Bengio et al. [4] for binarizing weights, and other variants of STE (e.g., in BNN) as approximate gradient.

as $\mathbf{w}^*$ for clarity. While our work easily extends to most discrete set $Q$, we focus on $Q = \{\pm 1\}^d$ since this binary setting remains most challenging and leads to the most significant savings. Existing binarization schemes can be largely divided into the following two categories.

**Post-Training Binarization (PTB):** we can formulate post-training binarization schemes as the following standard forward and backward pass:

$$\mathbf{w}_t = \mathbf{P}_Q(\mathbf{w}_t^*), \qquad \mathbf{w}_{t+1}^* = \mathbf{w}_t^* - \eta_t \widetilde{\nabla}\ell(\mathbf{w}_t^*),$$

where $\mathbf{P}_Q$ is the projector that binarizes the continuous weights $\mathbf{w}^*$ deterministically (e.g., the sign function) or stochastically[3], and $\widetilde{\nabla}\ell(\mathbf{w}_t^*)$ denotes a sample (sub)gradient of $\ell$ at $\mathbf{w}_t^*$. We point out that PTB is merely a post-processing step, i.e., the binarized weights $\mathbf{w}_t$ do not affect the update of the continuous weights $\mathbf{w}_t^*$, which are obtained through normal training. As a result, there is no guarantee that the acquired discrete weights $\mathbf{w}_t$ is a good solution (either global or local) to eq. (1).

**Binarization-Aware Training (BAT):** we then recall the more difficult binarization-aware training scheme BinaryConnect (BC), first initialized by Courbariaux et al. [11]:

$$\mathbf{w}_t = \mathbf{P}_Q(\mathbf{w}_t^*), \qquad \mathbf{w}_{t+1}^* = \mathbf{w}_t^* - \eta_t \widetilde{\nabla}\ell(\mathbf{w}_t), \qquad (2)$$

where we spot that the gradient is evaluated at the binarized weights $\mathbf{w}_t$ but used to update the continuous weights $\mathbf{w}_t^*$. This approach is also known as Straight Through Estimator [STE, 4]. Note that it is also possible to update the binarized weights instead, effectively performing the proximal gradient algorithm to solve (1), as shown by Bai et al. [2]:

$$\mathbf{w}_t = \mathbf{P}_Q(\mathbf{w}_t^*), \qquad \mathbf{w}_{t+1}^* = \mathbf{w}_t - \eta_t \widetilde{\nabla}\ell(\mathbf{w}_t).$$

This method is known as ProxQuant, and will serve as a baseline in our experiments.

## 2.1 ProxConnect

Dockhorn et al. [13] proposed ProxConnect (PC) as a broad generalization of BinaryConnect in (2):

$$\mathbf{w}_t = \mathbf{P}_{\mathsf{r}}^{\mu_t}(\mathbf{w}_t^*), \quad \mathbf{w}_{t+1}^* = \mathbf{w}_t^* - \eta_t \widetilde{\nabla}\ell(\mathbf{w}_t), \qquad (3)$$

where $\mu_t := 1 + \sum_{\tau=1}^{t-1} \eta_\tau$, $\eta_t > 0$ is the step size, and $\mathbf{P}_{\mathsf{r}}^{\mu_t}$ is the proximal quantizer:

$$\mathbf{P}_{\mathsf{r}}^{\mu}(\mathbf{w}) := \operatorname*{argmin}_{\mathbf{z}} \tfrac{1}{2\mu}\|\mathbf{w} - \mathbf{z}\|_2^2 + \mathsf{r}(\mathbf{z}), \text{ and}$$

$$\mathcal{M}_{\mathsf{r}}^{\mu}(\mathbf{w}) := \min_{\mathbf{z}} \tfrac{1}{2\mu}\|\mathbf{w} - \mathbf{z}\|_2^2 + \mathsf{r}(\mathbf{z}).$$

In particular, when the regularizer $\mathsf{r} = \iota_Q$ (the indicator function of $Q$), $\mathbf{P}_{\mathsf{r}}^{\mu_t} = \mathbf{P}_Q$ (for any $\mu_t$) and we recover BC in (2). Dockhorn et al. [13] showed that the PC update (3) amounts to applying the generalized conditional gradient algorithm to a smoothened dual of the regularized problem:

$$\min_{\mathbf{w}} \; [\ell(\mathbf{w}) + \mathsf{r}(\mathbf{w})] \approx \min_{\mathbf{w}^*} \; \ell^*(-\mathbf{w}^*) + \mathcal{M}_{\mathsf{r}^*}^{\mu}(\mathbf{w}^*),$$

where $f^*(\mathbf{w}^*) := \max_{\mathbf{w}}\langle \mathbf{w}, \mathbf{w}^* \rangle - f(\mathbf{w})$ is the Fenchel conjugate of $f$. The theory behind PC thus formally justifies STE from an optimization perspective. We provide a number of examples of the proximal quantizer $\mathbf{P}_{\mathsf{r}}^{\mu_t}$ in Appendix A.

Another natural cousin of PC is the reversed PC (rPC):

$$\mathbf{w}_t = \mathbf{P}_{\mathsf{r}}^{\mu_t}(\mathbf{w}_t^*), \quad \mathbf{w}_{t+1}^* = \mathbf{w}_t - \eta_t \widetilde{\nabla}\ell(\mathbf{w}_t^*),$$

which is able to exploit the rich landscape of the loss by evaluating the gradient at the continuous weights $\mathbf{w}_t^*$. Thus, we also include it as a baseline in our experiments.

We further discuss other related works in Appendix B.

---

[3]We only consider deterministic binarization in this paper.

## 3 Methodology

One popular heuristic to explain BC is through the following reformulation of problem (1):

$$\min_{\mathbf{w}^*} \ell\big(\mathbf{P}_Q(\mathbf{w}^*)\big).$$

Applying (stochastic) "gradient" to update the continuous weights we obtain:

$$\mathbf{w}_{t+1}^* = \mathbf{w}_t^* - \eta_t \cdot \mathbf{P}_Q'(\mathbf{w}_t^*) \cdot \widetilde{\nabla}\ell(\mathbf{P}_Q(\mathbf{w}_t^*)).$$

Unfortunately, the derivative of the projector $\mathbf{P}_Q$ is 0 everywhere except at the origin, where the derivative actually does not exist. BC [11], see (2), simply "pretended" that $\mathbf{P}_Q' = I$. Later works propose to replace the troublesome $\mathbf{P}_Q'$ by the derivative of functions that approximate $\mathbf{P}_Q$, e.g., the hard tanh in BNN [22] and the sign-Swish in BNN+ [12]. Despite their empirical success, it is not clear what is the underlying optimization problem or if it is possible to also replace the projector inside $\widetilde{\nabla}\ell$, i.e., allowing the algorithm to evaluate gradients at continuous weights, a clear advantage demonstrated by Bai et al. [2] and Dockhorn et al. [13]. Moreover, the theory established in PC, through a connection to the generalized conditional gradient algorithm, does not apply to these modifications yet, which is a gap that we aim to fill in this section.

### 3.1 ProxConnect++

To address the above-mentioned issues, we propose to study the following regularized problem:

$$\min_{\mathbf{w}^*} \ell(\mathsf{T}(\mathbf{w}^*)) + \mathsf{r}(\mathbf{w}^*), \tag{4}$$

as a relaxation of the (equivalent) reformulation of (1):

$$\min_{\mathbf{w}^*} \ell(\mathbf{P}_Q(\mathbf{w}^*)) + \iota_Q(\mathbf{w}^*).$$

In other words, $\mathsf{T} : \mathbb{R}^d \to \mathbb{R}^d$ is some transformation that approximates $\mathbf{P}_Q$ and the regularizer $\mathsf{r} : \mathbb{R}^d \to \mathbb{R}$ approximates the indicator function $\iota_Q$. Directly applying ProxConnect in (3) we obtain[4]:

$$\mathbf{w}_t = \mathbf{P}_\mathsf{r}^{\mu_t}(\mathbf{w}_t^*), \;\; \mathbf{w}_{t+1}^* = \mathbf{w}_t^* - \eta_t \mathsf{T}'(\mathbf{w}_t) \cdot \widetilde{\nabla}\ell\big(\mathsf{T}(\mathbf{w}_t)\big). \tag{5}$$

Introducing the forward and backward proximal quantizers:

$$\mathsf{F}_\mathsf{r}^\mu := \mathsf{T} \circ \mathbf{P}_\mathsf{r}^\mu, \quad \mathsf{B}_\mathsf{r}^\mu := \mathsf{T}' \circ \mathbf{P}_\mathsf{r}^\mu, \tag{6}$$

we can rewrite the update in (5) simply as:

$$\mathbf{w}_{t+1}^* = \mathbf{w}_t^* - \eta_t \cdot \mathsf{B}_\mathsf{r}^{\mu_t}(\mathbf{w}_t^*) \cdot \widetilde{\nabla}\ell\big(\mathsf{F}_\mathsf{r}^{\mu_t}(\mathbf{w}_t^*)\big). \tag{7}$$

It is clear that the original ProxConnect corresponds to the special choice

$$\mathsf{F}_\mathsf{r}^\mu = \mathbf{P}_\mathsf{r}^\mu, \quad \mathsf{B}_\mathsf{r}^\mu \equiv I.$$

Of course, one may now follow the recipe in (6) to design new forward-backward quantizers. We call this general formulation in (7) ProxConnect++ (PC++), which covers a broad family of algorithms.

Conversely, the complete characterization of proximal quantizers in Dockhorn et al. [13] allows us also to reverse engineer $\mathsf{T}$ and $\mathsf{r}$ from manually designed forward and backward quantizers. As we will see, most existing forward-backward quantizers turn out to be special cases of our proximal quantizers, and thus their empirical success can be justified from an optimization perspective. Indeed, for simplicity, let us restrict all quantizers to univariate ones that apply component-wise. Then, the following result is proven in Appendix C.

**Corollary 1.** *A pair of forward-backward quantizers* $(\mathsf{F}, \mathsf{B})$ *admits the decomposition in* (6) *(for some smoothing parameter* $\mu$ *and regularizer* $\mathsf{r}$) *iff both* $\mathsf{F}$ *and* $\mathsf{B}$ *are functions of* $\mathbf{P}(w) := \int_{-\infty}^w \frac{1}{\mathsf{B}(\omega)} \, d\mathsf{F}(\omega)$, *which is proximal (i.e., monotone, compact-valued and with a closed graph).*

---

[4]We assume throughout that $\mathsf{T}$, and any function whose derivative we use, are locally Lipschitz so that their generalized derivative is always defined, see Rockafellar and Wets [50].

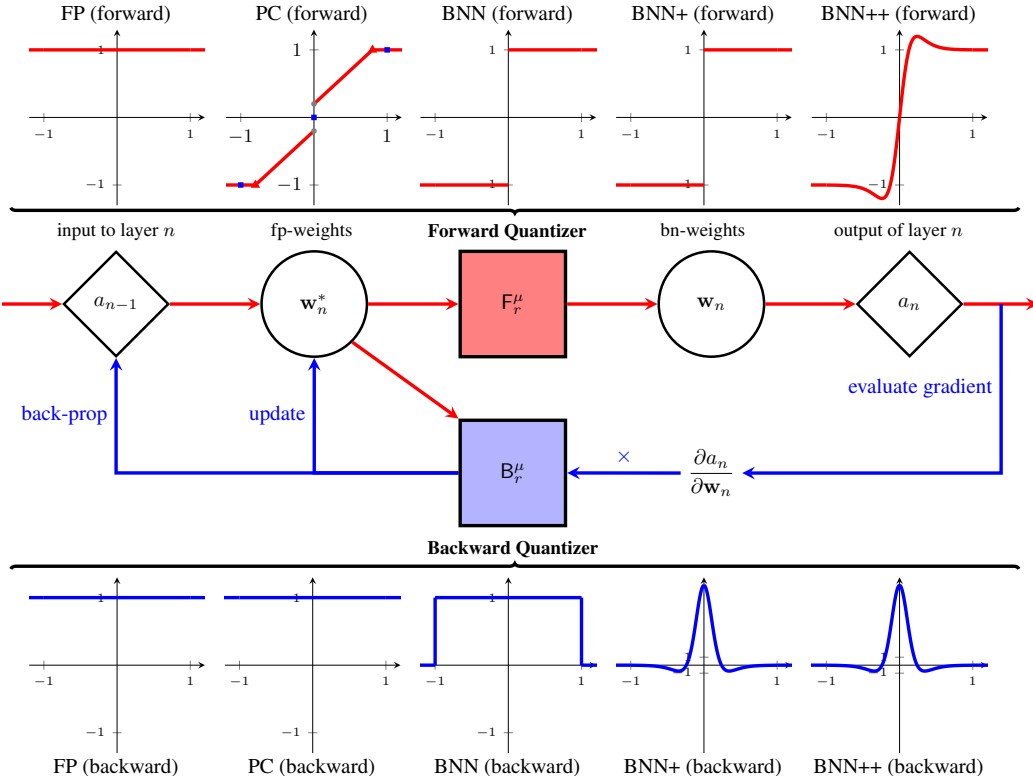

Figure 1: Forward and backward pass for ProxConnect++ algorithms (red/blue arrows indicate the forward/backward pass), where fp denotes full precision, bn denotes binary and back-prop denotes backpropagation.

Importantly, with forward-backward proximal quantizers, the convergence results established by Dockhorn et al. [13] for PC directly carries over to PC++ (see Appendix C for details). Let us further illustrate the convenience of Corollary 1 by some examples.

**Example 1** (BNN). *Hubara et al. [22] proposed BNN with the choice*

$$\mathsf{F} = \text{sign} \quad and \quad \mathsf{B} = \mathbf{1}_{[-1,1]},$$

*which satisfies the decomposition in* (6). *Indeed, let*

$$\mathsf{T}(w) = \min\{1, \max\{-1, w\}\}, \tag{8}$$

$$\mathbf{P}_r^\mu(w) = \begin{cases} \frac{1}{\mu}w + \text{sign}(w)(1 - \frac{1}{\mu}), & \text{if } |w| > 1 \\ \text{sign}(w), & \text{if } |w| \le 1 \end{cases}. \tag{9}$$

*Since* $\mathsf{B}$ *is constant over* $[-1, 1]$, *applying Corollary 1 we deduce that the proximal quantizer* $\mathbf{P}_r^\mu$, *if exists, must coincide with* $\mathsf{F}$ *over the support of* $\mathsf{B}$. *Applying monotonicity of* $\mathbf{P}_r^\mu$ *we may complete the reverse engineering by making the choice over* $|w| > 1$ *as indicated above. We can easily verify the decomposition in* (6):

$$\mathsf{F} = \text{sign} = \mathsf{T} \circ \mathbf{P}_r^\mu, \quad \mathsf{B} = \mathbf{1}_{[-1,1]} = \mathsf{T}' \circ \mathbf{P}_r^\mu.$$

*Thus, BNN is exactly BinaryConnect applied to the transformed problem in* (4), *where the transformation* $\mathsf{T}$ *is the so-called hard tanh in* (8) *while the regularizer* r *is determined (implicitly) by the proximal quantizer* $\mathbf{P}_r^\mu$ *in* (9).

To our best knowledge, this is the first time the (regularized) objective function that BNN aims to optimize has been identified. The convergence properties of BNN hence follow from the general result of Dockhorn et al. [13] on ProxConnect, see Appendix C.

Table 1: Variants of ProxConnect++.

| Forward Quantizer | Backward Quantizer | Algorithm |
|---|---|---|
| identity | identity | FP |
| $\mathbf{P}_Q$ | identity | BC |
| $\mathbf{L}_\rho^\ell$ | identity | PC |
| $\mathbf{P}_Q$ | $\mathbf{1}_{[-1,1]}$ | BNN |
| $\mathbf{P}_Q$ | $\nabla$SS | BNN+ |
| SS | $\nabla$SS | BNN++ |

**Example 2** (BNN+). *Darabi et al. [12] adopted the derivative of the sign-Swish (SS) function as a backward quantizer while retaining the sign function as the forward quantizer:*

$$\mathsf{B}(\mathbf{w}) = \nabla\mathrm{SS}(\mathbf{w}) := \mu[1 - \tfrac{\mu\mathbf{w}}{2}\tanh(\tfrac{\mu\mathbf{w}}{2})]\tanh'(\tfrac{\mu\mathbf{w}}{2}), \ \ \mathsf{F} = \mathrm{sign},$$

*where $\mu$ is a hyperparameter that controls how well SS approximates the sign. Applying Corollary 1 we find that the derivative of SS (as backward) coupled with the sign (as forward) do not admit the decomposition in (6), for any regularizer r. Thus, we are not able to find the (regularized) objective function (if it exists) underlying BNN+.*

We conclude that BNN+ cannot be justified under the framework of PC++. However, it is possible to design a variant of BNN+ that does belong to the PC++ family and hence enjoys the accompanying theoretical properties:

**Example 3** (BNN++). *We propose that a simple fix of BNN+ would be to replace its* sign *forward quantizer with the sign-Swish (SS) function:*

$$\mathsf{F}(\mathbf{w}) = \mathrm{SS}(\mathbf{w}) := \tfrac{\mu\mathbf{w}}{2}\tanh'(\tfrac{\mu\mathbf{w}}{2}) + \tanh(\tfrac{\mu\mathbf{w}}{2}),$$

*which is simply the primitive of $\mathsf{B}$. In this case, the algorithm simply reduces to PC++ applied on (4) with r $= 0$ (and hence essentially stochastic gradient descent). Of course, we could also compose with a proximal quantizer to arrive at the pair $(\mathsf{F} \circ \mathbf{P}_r^\mu, \mathsf{B} \circ \mathbf{P}_r^\mu)$, which effectively reduces to PC++ applied on the regularized objective in (4) with a nontrivial r. We call this variant BNN++.*

We will demonstrate in the next section that BNN++ is more desirable than BNN+ empirically.

More generally, we have the following result on designing new forward-backward quantizers:

**Corollary 2.** *If the forward quantizer is continuously differentiable (with bounded support), then one can simply choose the backward quantizer as the derivative of the forward quantizer.*

This follows from Corollary 1 since $\mathsf{P}(w) \equiv w$ is clearly proximal. Note that the BNN example does not follow from Corollary 2. In Appendix F, we provide additional examples of forward-backward quantizers based on existing methods, and we show that Corollary 2 consistently improves previous practices.

In summary: (1) ProxConnect++ enables us to design forward-backward quantizers with infinite many choices of T and r, (2) it also allows us to reverse engineer T and r from existing forward-backward quantizers, which helps us to better understand existing practices, (3) with our theoretical tool, we design a new BNN++ algorithm, which enjoys immediate convergence properties. Figure 1 visualizes ProxConnect++ with a variety of forward-backward quantizers.

## 4 Experiments

In this section, we perform extensive experiments to benchmark PC++ on CNN backbone models and the recently advanced vision transformer architectures in three settings: (a) binarizing weights only (BW); (b) binarizing weights and activations (BWA), where we simply apply a similar forward-backward proximal quantizer to the activations; and (c) binarizing weights, activations, with 8-bit dot-product accumulators (BWAA) [43].

Table 2: Binarizing weights (BW), binarizing weights and activation (BWA) and binarizing weights, activation, with 8-bit accumulators (BWAA) on CNN backbones. We consider the fine-tuning (FT) pipeline and the end-to-end (E2E) pipeline. We compare five variants of ProxConnect++ (BC, PC, BNN, BNN+, and BNN++) with FP, PQ, and rPC in terms of test accuracy. For the end-to-end pipeline, we omit the results for BWAA due to training divergence and report the mean of five runs with different random seeds.

| Dataset | Pipeline | Task | FP | PQ | rPC | ProxConnect++ | | | | |
| --- | --- | --- | --- | --- | --- | --- | --- | --- | --- | --- |
| | | | | | | BC | PC | BNN | BNN+ | BNN++ |
| CIFAR-10 | FT | BW | 92.01% | 89.94% | 89.98% | 90.31% | 90.31% | 90.35% | 90.27% | **90.40%** |
| | | BWA | 92.01% | 88.79% | 83.55% | 89.39% | 89.95% | 90.01% | 89.99% | **90.22%** |
| | | BWAA | 92.01% | 85.39% | 81.10% | 89.11% | 89.21% | 89.32% | 89.55% | **90.01%** |
| | E2E | BW | 92.01% | 81.59% | 81.82% | 87.51% | 88.05% | 89.92% | 89.39% | **90.03%** |
| | | BWA | 92.01% | 81.51% | 81.60% | 86.99% | 87.26% | 89.15% | 89.02% | **89.91%** |
| ImageNet-1K | FT | BW | 78.87% | 66.77% | 69.22% | 71.35% | 71.29% | 71.41% | 70.22% | **72.33%** |
| | | BWA | 78.87% | 56.21% | 58.19% | 65.99% | 65.61% | 66.02% | 65.22% | **68.03%** |
| | | BWAA | 78.87% | 53.29% | 55.28% | 58.18% | 59.21% | 59.77% | 59.10% | **63.02%** |
| | E2E | BW | 78.87% | 63.23% | 66.39% | 67.45% | 67.51% | 67.49% | 66.99% | **68.11%** |
| | | BWA | 78.87% | 61.19% | 64.17% | 65.42% | 65.31% | 65.29% | 65.98% | **66.08%** |

## 4.1 Experimental settings

**Datasets**: We perform image classification on CIFAR-10/100 datasets [27] and ImageNet-1K dataset [28]. Additional details on our experimental setting can be found in Appendix D.

**Backbone architectures:** (1) *CNNs*: we evaluate CIFAR-10 classification using ResNet20 [18], and ImageNet-1K with ResNet-50 [18]. We consider both fine-tuning and end-to-end training; (2) *Vision transformers*: we further evaluate our algorithm on two popular vision transformer models: ViT [14] and DeiT [52]. For ViT, we consider ViT-B model and fine-tuning task across all models[5]. For DeiT, we consider DeiT-B, DeiT-S, and DeiT-T, which consist of 12, 6, 3 building blocks and 768, 384 and 192 embedding dimensions, respectively; we consider fine-tuning task on ImageNet-1K pre-trained model for CIFAR datasets and end-to-end training on ImageNet-1K dataset.

**Baselines**: For ProxConnect++, we consider the 6 variants in Table 1. With different choices of the forward quantizer $F_r^\mu$ and the backward quantizer $B_r^\mu$, we include the full precision (FP) baseline and 5 binarization methods: BinaryConnect (BC) [11], ProxConnect (PC) [13], Binary Neural Network (BNN) [22], the original BNN+ [12], and the modified BNN++ with $F_r^\mu = SS$. Note that we linearly increase $\mu$ in BNN++ to achieve full binarization in the end. We also compare ProxConnect++ with the ProxQuant and reverseProxConnect baselines.

**Hyperparameters**: We apply the same training hyperparameters and fine-tune/end-to-end training for 100/300 epochs across all models. For binarization methods: (1) PQ (ProxQuant): similar to Bai et al. [2], we apply the LinearQuantizer (LQ), see (10) in Appendix A, with initial $\rho_0 = 0.01$ and linearly increase to $\rho_T = 10$; (2) rPC (reverseProxConnect): we use the same LQ for rPC; (3) ProxConnect++: for PC, we apply the same LQ; for BNN+, we choose $\mu = 5$ (no need to increase $\mu$ as the forward quantizer is sign); for BNN++, we choose $\mu_0 = 5$ and linearly increase to $\mu_T = 30$ to achieve binarization at the final step.

Across all the experiments with random initialization, we report the mean of three runs with different random seeds. Furthermore, we provide the complete results with error bars in Appendix G.

## 4.2 CNN as backbone

We first compare PC++ against baseline methods on various tasks employing CNNs:

(1) Binarizing weights only (BW), where we simply binarize the weights and keep the other components (i.e., activations and accumulations) in full precision.

---

[5]Note that we use pre-trained models provided by Dosovitskiy et al. [14] on the ImageNet-21K/ImageNet-1K for fine-tuning ViT-B model on the ImageNet-1K/CIFAR datasets, respectively.

Table 3: Our results on binarizing vision transformers (binarizing weights only). We compare five variants of ProxConnect++ (BC, PC, BNN, BNN+, and BNN++) with FP, PQ, and rPC. End-to-end training tasks are marked as **bold** (i.e., ImageNet-1K for DeiT-T/S/B), where the results are the mean of five runs with different random seeds.

| Model | Dataset | FP | PQ | rPC | ProxConnect++ | | | | |
| --- | --- | --- | --- | --- | --- | --- | --- | --- | --- |
| | | | | | BC | PC | BNN | BNN+ | BNN++ |
| | CIFAR-10 | 98.13% | 85.06% | 86.22% | 87.97% | 90.12% | 89.08% | 88.12% | **90.24%** |
| ViT-B | CIFAR-100 | 87.14% | 72.07% | 73.52% | 76.35% | 78.13% | 77.23% | 77.10% | **79.22%** |
| | ImageNet-1K | 77.91% | 57.65% | 55.33% | 63.24% | **66.33%** | 65.31% | 63.55% | **66.33%** |
| | CIFAR-10 | 94.86% | 82.76% | 82.25% | 83.10% | 85.15% | 86.12% | 85.91% | **86.41%** |
| DeiT-T | CIFAR-100 | 72.37% | 54.55% | 55.66% | 59.65% | 60.15% | 60.06% | 59.77% | **60.33%** |
| | **ImageNet-1K** | 72.20% | 61.23% | 60.35% | 63.22% | 66.15% | 65.00% | 66.67% | **67.34%** |
| | CIFAR-10 | 95.10% | 81.67% | 80.23% | 84.85% | 85.13% | 85.09% | 85.16% | **86.19%** |
| DeiT-S | CIFAR-100 | 73.19% | 45.55% | 46.66% | 60.12% | 61.59% | 60.55% | 60.17% | **62.98%** |
| | **ImageNet-1K** | 79.91% | 69.87% | 68.74% | 73.16% | 73.51% | 73.77% | 73.23% | **73.53%** |
| | CIFAR-10 | 98.72% | 85.22% | 86.35% | 88.95% | 90.53% | 90.21% | 89.03% | **90.67%** |
| DeiT-B | CIFAR-100 | 86.65% | 72.11% | 73.40% | 75.40% | **78.55%** | 76.22% | 76.51% | 78.30% |
| | **ImageNet-1K** | 81.81% | 72.54% | 70.11% | 76.55% | 76.61% | 75.60% | 76.63% | **76.74%** |

Table 4: Results on binarizing vision transformers (BW, BWA, and BWAA) on DeiT-T. We compare 5 variants of ProxConnect++ (BC, PC, BNN, BNN+, and BNN++) with FP, PQ, and rPC. End-to-end training tasks are marked as **bold** (i.e., ImageNet-1K), where we omit the results for BWAA due to training divergence and the reported results are the mean of five runs with different random seeds.

| Dataset | Task | FP | PQ | rPC | ProxConnect++ | | | | |
| --- | --- | --- | --- | --- | --- | --- | --- | --- | --- |
| | | | | | BC | PC | BNN | BNN+ | BNN++ |
| | BW | 94.85% | 82.76% | 82.25% | 83.10% | 85.15% | 86.12% | 85.91% | **86.41%** |
| CIFAR-10 | BWA | 94.85% | 82.56% | 82.02% | 82.89% | 85.01% | 85.99% | 85.66% | **86.12%** |
| | BWAA | 94.85% | 81.34% | 80.97% | 82.08% | 84.31% | 84.87% | 84.72% | **85.31%** |
| | BW | 72.37% | 54.55% | 55.66% | 59.65% | 60.15% | 60.06% | 59.77% | **60.33%** |
| CIFAR-100 | BWA | 72.37% | 53.77% | 54.98% | 59.21% | 59.71% | 59.66% | 59.12% | **59.85%** |
| | BWAA | 72.37% | 52.15% | 54.36% | 58.15% | 59.01% | 58.72% | 58.15% | **59.06%** |
| **ImageNet-1K** | BW | 72.20% | 61.23% | 60.35% | 63.23% | 66.15% | 65.00% | 66.67% | **67.34%** |
| | BWA | 72.20% | 60.01% | 58.77% | 62.13% | 65.29% | 63.75% | 65.29% | **65.65%** |

(2) Binarizing weights and activations (BWA), while keeping accumulation in full precision. Similar to the weights, we apply the same forward-backward proximal quantizer to binarize activations.

(3) Binarizing weights, activations, with 8-bit accumulators (BWAA). BWAA is more desirable in certain cases where the network bandwidth is narrow, e.g., in homomorphic encryption. To achieve BWAA, in addition to quantizing the weights and activations, we follow the implementation of WrapNet [43] and quantize the accumulation of each layer with an additional cyclic function. In practice, we find that with 1-bit weights and activations, the lowest bits we can successfully employ to quantize accumulation is 8, while any smaller choice would raise a high overflow rate and cause the network to diverge. Moreover, BWAA highly relies on a good initialization and cannot be successfully trained end-to-end in our evaluation (and hence omitted).

Note that for the fine-tuning pipeline, we initialize the model with their corresponding pre-trained full precision weights. For the end-to-end pipeline, we utilize random initialization. We report our results in Table 2 and observe: (1) the PC family outperforms baseline methods (i.e., PQ and rPC), and achieves competitive performance on both small and larger scale datasets; (2) BNN++ performs consistently better and is more desirable among the five variants of PC++, especially on BWA and BWAA tasks. Its advantage over BNN+ further validates our theoretical guidance.

## 4.3 Vision transformer as backbone

Next, we perform similar experiments on the three tasks on vision transformers.

Table 5: Ablation study on the effect of the scaling factor, normalization, pre-training, and knowledge distillation. Experiments are performed on CIFAR-10 with ViT-B.

| Method | Scaling | Normalization | Pre-train | KD | Accuracy |
|--------|---------|---------------|-----------|-----|----------|
| PC | ✗ | ✗ | ✗ | ✗ | 0.10% |
| | ✓ | ✗ | ✗ | ✗ | 12.81% |
| | ✓ | ✓ | ✗ | ✗ | 66.51% |
| | ✓ | ✓ | ✓ | ✗ | 88.53% |
| | ✓ | ✓ | ✓ | ✓ | 90.13% |
| BNN++ | ✗ | ✗ | ✗ | ✗ | 1.50% |
| | ✓ | ✗ | ✗ | ✗ | 23.55% |
| | ✓ | ✓ | ✗ | ✗ | 77.22% |
| | ✓ | ✓ | ✓ | ✗ | 89.05% |
| | ✓ | ✓ | ✓ | ✓ | 90.22% |

**Implementation on vision transformers**: While network binarization is popular for CNNs, its application for vision transformers is still rare[6]. Here we apply four protocols for implementation:

(1) We keep the mean $s_n$ of full precision weights $\mathbf{w}_n^*$ for each layer $n$ as a scaling factor (can be thus absorbed into $\mathsf{F}_r^{\mu_t}$) for the binary weights $\mathbf{w}_n$. Such an approach keeps the range of $\mathbf{w}_n^*$ during binarization and significantly reduces training difficulty without additional computation.

(2) For binarized vision transformer models, LayerNorm is important to avoid gradient explosion. Thus, we add one more LayerNorm layer at the end of each attention block.

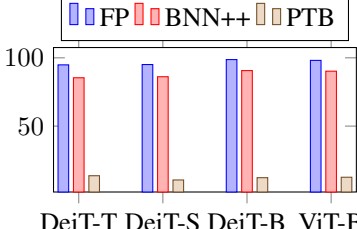

Figure 2: Comparison between Full Precision (FP) model, BNN++, and Post-training Binarization (PTB) on the fine-tuning task on CIFAR-10.

(3) When fine-tuning a pre-trained model (full precision), the binarized vision transformer usually suffers from a bad initialization. Thus, a few epochs of pre-training on the binarized vision transformer is extremely helpful and can make fine-tuning much more efficient and effective.

(4) We apply the knowledge distillation technique in BiBERT [45] to boost the performance. We use full precision pre-trained models as the teacher model.

**Main Results**: We report the main results of binarizing vision transformers in Table 3 (BW) and Table 4 (BW, BWA, BWAA), where we compare ProxConnect++ algorithms with the FP, PQ, and rPC baselines on fine-tuning and end-to-end training tasks. We observe that: (1) ProxConnect++ variants generally outperform PQ and rPC and are able to binarize vision transformers with less than $10\%$ accuracy degradation on the BW task. In particular, for end-to-end training, the best performing ProxConnect++ algorithms achieve $\approx 5\%$ accuracy drop; (2) Among the five variants, we confirm BNN++ is also generally better overall for vision transformers. This provides evidence that our Corollary 1 allows practitioners to easily design many and choose the one that performs best empirically; (3) With a clear underlying optimization objective, BNN++ again outperforms BNN+ across all tasks, which empirically verifies our theoretical findings on vision transformers; (4) In general, we find that weight binarization achieves about 30x reduction in memory footprint, e.g., from 450 MB to 15 MB for ViT-B.

**Ablation Studies**: We provide further ablation studies to gain more insights and verify our binarization protocols for vision transformers.

*Post-training Binarization*: in Figure 2, we verify the difference between PTB (post-training binarization) and BAT (binarization-aware training) on the fine-tuning task on CIFAR-10 across different models. Note that we use BNN++ as a demonstration of BAT. We observe that without optimization during fine-tuning, the PTB approach fails in general, thus confirming the importance of considering BAT for vision transformers.

*Effect of binarizing protocols*: here we show the effect of the four binarizing protocols mentioned at the beginning, including scaling the binarized weights using the mean of full precision weights

---

[6]Notably, He et al. [19] also consider binarizing vision transformers, which we compare our implementation details and experimental results against in Appendix E.

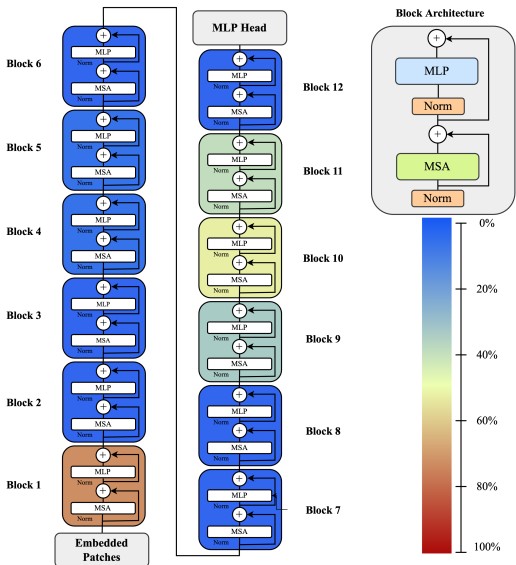

Figure 3: Results of binarizing different components (blocks) of ViT-B architecture on CIFAR-10. Warmer color indicates significant accuracy degradation after binarization.

(scaling), adding additional LayerNorm layers (normalization), BAT on the full precision pre-trained models (pre-train) and knowledge distillation. We report the results in Table 5 and confirm that each protocol is essential to binarize vision transformers successfully.

*Which block should one binarize*: lastly, we visualize the sensitivity of each building block to binarization in vision transformers (i.e., ViT-B) on CIFAR-10 in Figure 3. We observe that binarizing blocks near the head and the tail of the architecture causes a significant accuracy drop.

# 5   Conclusion

In this work we study the popular *approximate gradient* approach in neural network binarization. By generalizing ProxConnect and proposing PC++, we provide a principled way to understand forward-backward quantizers and cover most existing binarization techniques as special cases. Furthermore, PC++ enables us to easily design the desired quantizers (e.g., the new BNN++) with automatic theoretical guarantees. We apply PC++ to CNNs and vision transformers and compare its variants in extensive experiments. We confirm empirically that PC++ overall achieves competitive results, whereas BNN++ is generally more desirable.

## Broader impacts and limitations

We anticipate our work to further enable training and deploying advanced machine learning models to resource limited devices and environments, and help reducing energy consumption and carbon footprint at large. We do not foresee any direct negative societal impact. One limitation we hope to address in the future is to build a theoretical framework that will allow practitioners to quickly evaluate different forward-backward quantizers for a variety of applications.

## Acknowledgments and Disclosure of Funding

We thank the reviewers and the area chair for thoughtful comments that have improved our final draft. We thank Arash Ardakani, Ali Mosleh and Marzieh Tahaei for their early participation in this project. YY gratefully acknowledges NSERC and CIFAR for funding support.

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

# Appendix for *Understanding Neural Network Binarization with Forward and Backward Proximal Quantizers*

## A  More on Proximal Quantizers

Dockhorn et al. [13] gave a complete characterization of the proximal quantizer $\mathbf{P}_r$: a (multi-valued) mapping $\mathbf{P}$ is a proximal quantizer (of some underlying regularizer r) iff it is monotone, compact-valued and with a closed graph. We now give a few examples to illustrate the ubiquity of proximal quantizers, as well as the generality of PC:

- Identity function: apparently, choosing $\mathbf{P}_r^{\mu_t}$ as the identity function recovers the full precision training.
- $\mathbf{P}_r^{\mu_t} = \mathbf{P}_Q$: as $Q = \{\pm 1\}$, this choice recovers exactly BC in (2).
- $\mathbf{P}_r^{\mu_t} = \mathbf{L}_\rho^\varrho$: This is the general piecewise linear quantizer designed by Dockhorn et al. [13]. Recall that $Q = \{q_k\}_{k=1}^2$, where $q_1 = -1, q_2 = +1$, such that $p_2 = 0$ is the middle point. By introducing two parameters $\rho, \varrho \geq 0$, we can define two shifts:

$$\text{horizontal:} \quad q_1^- = q_1, q_1^+ = p_2 \wedge (q_1 + \rho)$$
$$q_2^- = p_2 \vee (q_2 - \rho), q_2^+ = q_2$$
$$\text{vertical:} \quad p_2^- = q_1 \vee (p_2 - \varrho), p_2^+ = q_2 \wedge (p_2 + \varrho).$$

Then, we define $\mathbf{L}_\rho^\varrho$ as the piece-wise linear map (that simply connects the points by straight lines):

$$\mathbf{L}_\rho^\varrho(w^*) = \begin{cases} q_1, & \text{if } q_1^- \leq w^* \leq q_1^+ \\ q_1 + (w^* - q_1^+)\frac{p_2^- - q_1}{p_2 - q_1^+}, & \text{if } q_1^+ \leq w^* < p_2 \\ p_2^+ + (w^* - p_2)\frac{q_2 - p_2^+}{q_2^- - p_2}, & \text{if } p_2 < w^* \leq q_2^- \\ q_2, & \text{if } q_2^- \leq w^* \leq q_2^+ \end{cases} \tag{10}$$

For the middle points, $\mathbf{L}_\rho^\varrho(w^*)$ can be regarded as the intermediate state between the identity function and $\mathbf{P}_Q$ such that, where $\mathbf{L}_\rho^\varrho(w^*)$ may take any value within the two limits. Note that $\rho$ controls the discretization vicinity, such that in practice, $\rho$ is linearly increased over time to fulfill binary weights in the end. We visualize examples of $\mathbf{L}_\rho^\varrho(w^*)$ in Figure 4.

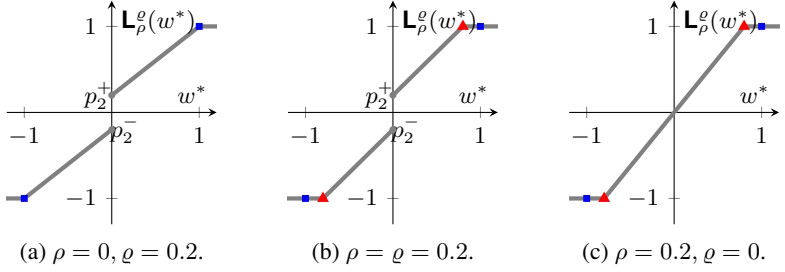

(a) $\rho = 0, \varrho = 0.2$.     (b) $\rho = \varrho = 0.2$.     (c) $\rho = 0.2, \varrho = 0$.

Figure 4: Different instantiations of the proximal map $\mathbf{L}_\rho^\varrho$ in (10) for $Q = \{-1, 1\}$.

## B  Related works

**Vision Transformer.** In computer vision, vision transformers have become one of the most popular backbone architectures. Dosovitskiy et al. [14] is the first to modify the transformer model to enable images as input, namely the ViT model. Specifically, Dosovitskiy et al. [14] translates an image to a sequence of flattened image patches as input, and applies a self-attention mechanism to retrieve patch-wise information in the feature representation. Touvron et al. [52] further equips ViT with knowledge distillation and proposes DeiT that generalizes well on smaller models and datasets. Liu et al. [34] further proposes Swin as a hierarchical vision transformer that computes representation with shifted windows.

In vision transformers, the main computation overhead is the multi-head attention layer, whose cost is quadratic with the length of the image patches. As a result, such models are in general expensive to train. To reduce the computational cost, different compression techniques have been explored. For instance, Pan et al. [44] performs dynamic pruning for less important patches; Bhojanapalli et al. [6] reuses attention scores computed for one layer in multiple building blocks; Hou and Kung [21] applies multi-dimensional model compression. In this paper, we focus on an alternative approach, namely network quantization.

**Network Quantization.**    We consider two possible scenarios of network quantization:

(1) Post-training Quantization: We first discuss the easier post-training quantization methods. Such approaches usually quantize the full-precision pre-trained model and directly apply it for inference. Post-training quantization is widely used in CNNs [1, 3, 9, 29, 41, 42, 54]. Liu et al. [38] is the first to explore PTQ for vision transformers. It optimizes the quantization intervals and considers ranking information in the loss function. However, it only considers quantization to a 6-bit model without severe performance degradation. For lower-bit quantization, it is essential to leverage training.

(2) Quantization-Aware Training (QAT): different from post-training quantization, quantization-aware training leverage quantization during pre-training or fine-tuning. Thus it can be formulated as an optimization problem for learning the optimal quantized weights [5, 10, 15, 17, 23–25, 39, 58]. Compared with PTQ, QAT can obtain less accuracy drop in low-bit quantization compared to the full-precision model. Li et al. [30] and Xu et al. [55] demonstrate that QAT requires a unique design to quantize vision transformers and it is possible to perform quantization to 3 bit without severe performance degradation. [19] further performs binarization with softmax-aware binarization and information preservation.

**Binarization Techniques:**    (1) Here we summarize existing binarization approaches that can be either justified or improved by PC++: Some existing implementations [31, 37, 46] set the forward quantizer as sign and design the backward quantizer in an ad hoc fashion (based on graphic approximation of the sign function), e.g., Liu et al. [37] applies a piece-wise polynomial approximation; Lin et al. [31] improves [37] with a dynamic polynomial approximation; Qin et al. [46] designs a dynamic error decay estimator based on the tanh function. These methods cannot be justified with PC++, but could be improved by designing new forward quantizers using our Corollary 2. We discuss these new variants in Appendix F and compare them with our methods. Other implementations [36, 53] applies shift transformation on both forward and backward quantizers, which belongs to the PC++ family.

(2) Architecture design that can be further integrated into our PC++ as future work: Xu et al. [56] designs a rectified clamp unit to address "dead weights"; Rastegari et al. [48] applies the absolute mean of weights and activations; Martinez et al. [40] uses real-to-binary attention matching and data-driven channel re-scaling; Kim et al. [26] proposes a shifted activation function.

(3) Optimization refinement, again could be integrated into our framework as future work: Liu et al. [33] utilizes state-aware gradient state; Liu et al. [35] provides a weight decay scheme; and Helwegen et al. [20] proposes Bop, a new optimizer for BNNs.

## C   Additional Theoretical Results

**Corollary 1.** *A pair of forward-backward quantizers* $(\mathsf{F}, \mathsf{B})$ *admits the decomposition in* (6) *(for some smoothing parameter $\mu$ and regularizer* r*) iff both* $\mathsf{F}$ *and* $\mathsf{B}$ *are functions of* $\mathbf{P}(w) := \int_{-\infty}^{w} \frac{1}{\mathsf{B}(\omega)} \, d\mathsf{F}(\omega)$*, which is proximal (i.e., monotone, compact-valued and with a closed graph).*

*Proof.*  We first recall the decomposition in (6):
$$\mathsf{F}_{\mathsf{r}}^{\mu} := \mathsf{T} \circ \mathbf{P}_{\mathsf{r}}^{\mu}, \quad \mathsf{B}_{\mathsf{r}}^{\mu} := \mathsf{T}' \circ \mathbf{P}_{\mathsf{r}}^{\mu}. \tag{19}$$
Suppose first that $(\mathsf{F}, \mathsf{B})$ satisfies the above decomposition. Clearly, both $\mathsf{F}$ and $\mathsf{B}$ are functions of $\mathbf{P} = \mathbf{P}_{\mathsf{r}}^{\mu}$. Moreover,
$$\frac{\mathsf{F}'(\omega)}{\mathsf{B}(\omega)} = \frac{\mathbf{P}_{\mathsf{r}}^{\mu'}(\omega) \cdot \mathsf{T}' \circ \mathbf{P}_{\mathsf{r}}^{\mu}}{\mathsf{T}' \circ \mathbf{P}_{\mathsf{r}}^{\mu}} = \mathbf{P}_{\mathsf{r}}^{\mu'}(\omega)$$

and thus

$$\int_{-\infty}^{w} \frac{1}{\mathsf{B}(\omega)} \, d\mathsf{F}(\omega) = \mathbf{P}_{\mathsf{r}}^{\mu}(w) - \mathbf{P}_{\mathsf{r}}^{\mu}(-\infty),$$

which is clearly proximal.

Conversely, let $\mathbf{P}(w) := \int_{-\infty}^{w} \frac{1}{\mathsf{B}(\omega)} \, d\mathsf{F}(\omega)$ be proximal. Taking (generalized) derivative we obtain

$$\mathbf{P}'(\omega) = \frac{\mathsf{F}'(\omega)}{\mathsf{B}(\omega)}.$$

Since B is a function of $\mathbf{P}$, say $\mathsf{B} = \mathsf{T}' \circ \mathbf{P}$, performing integration we obtain

$$\mathsf{F} = \mathsf{T} \circ \mathbf{P},$$

up to some immaterial constant (that can be absorbed into T). Thus, $(\mathsf{F}, \mathsf{B})$ satisfies the decomposition (6). $\qquad\square$

The following convergence guarantee for PC++ follows directly from the results in Dockhorn et al. [13]:

**Theorem 1.** *Fix any* $\mathbf{w}$*, the iterates in* (7) *satisfy:*

$$\sum_{\tau=s}^{t} \eta_\tau [\langle \mathbf{w}_\tau - \mathbf{w}, \widetilde{\nabla}\ell(\mathsf{T}\mathbf{w}_\tau)\rangle + \mathsf{r}(\mathbf{w}_\tau) - \mathsf{r}(\mathbf{w})] \le \Delta_{s-1}(\mathbf{w}) - \Delta_t(\mathbf{w}) + \sum_{\tau=s}^{t} \Delta_\tau(\mathbf{w}_\tau), \quad (11)$$

*where* $\Delta_\tau(\mathbf{w}) := \mathsf{r}_\tau(\mathbf{w}) - \mathsf{r}_\tau(\mathbf{w}_{\tau+1}) - \langle \mathbf{w} - \mathbf{w}_{\tau+1}, \mathbf{w}_{\tau+1}^* \rangle$ *is the Bregman divergence induced by the (possibly nonconvex) function* $\mathsf{r}_\tau(\mathbf{w}) := \mu_{\tau+1} \cdot \mathsf{r}(\mathbf{w}) + \frac{1}{2}\|\mathbf{w}\|_2^2$*. (Recall that* $\mu_t := 1 + \sum_{\tau=1}^{t-1} \eta_\tau$*.)*

The summand on the left-hand side of (11) is related to the duality gap, which is a natural measure of stationarity for the nonconvex problem (4). Indeed, it reduces to the familiar ones when convexity is present:

**Theorem 2.** *For convex* $\ell \circ \mathsf{T}$ *and any* $\mathbf{w}$*, the iterates in* (7) *satisfy:*

$$\min_{\tau=s,\dots,t} \mathbb{E}[f(\mathbf{w}_\tau) - f(\mathbf{w})] \le \frac{1}{\sum_{\tau=s}^{t} \eta_\tau} \cdot \mathbb{E}\Big[\Delta_{s-1}(\mathbf{w}) - \Delta_t(\mathbf{w}) + \sum_{\tau=s}^{t} \Delta_\tau(\mathbf{w}_\tau)\Big]. \quad (12)$$

*If* r *is also convex, then*

$$\min_{\tau=s,\dots,t} \mathbb{E}[f(\mathbf{w}_\tau) - f(\mathbf{w})] \le \frac{1}{\sum_{\tau=s}^{t} \eta_\tau} \cdot \mathbb{E}\Big[\Delta_{s-1}(\mathbf{w}) + \sum_{\tau=s}^{t} \frac{\eta_\tau^2}{2}\|\widetilde{\nabla}\ell(\mathbf{w}_\tau)\|_2^2\Big], \quad (13)$$

*and*

$$\mathbb{E}\big[f(\bar{\mathbf{w}}_t) - f(\mathbf{w})\big] \le \frac{1}{\sum_{\tau=s}^{t} \eta_\tau} \cdot \mathbb{E}\Big[\Delta_{s-1}(\mathbf{w}) + \sum_{\tau=s}^{t} \frac{\eta_\tau^2}{2}\|\widetilde{\nabla}\ell(\mathbf{w}_\tau)\|_2^2\Big], \quad (14)$$

*where* $\mathbf{w}_t = \frac{\sum_{\tau=s}^{t} \eta_\tau \mathbf{w}_\tau}{\sum_{\tau=s}^{t} \eta_\tau}$*, and* $f := \ell \circ \mathsf{T} + \mathsf{r}$ *is the regularized and transformed objective.*

The right-hand sides of (13) and (14) diminish iff $\eta_t \to 0$ and $\sum_t \eta_t = \infty$ (assuming boundedness of the stochastic gradient). We note some trade-off in choosing the step size $\eta_\tau$: both the numerator and denominator of the right-hand sides of (13) and (14) are increasing w.r.t. $\eta_\tau$. The same conclusion can be drawn for (12) and (11), where $\Delta_\tau$ also depends on $\eta_\tau$ (through the accumulated magnitude of $\mathbf{w}_{\tau+1}^*$).

## D   Additional Experimental Settings

**Hardware and package:**   All experiments were run on a GPU cluster with `NVIDIA V100` GPUs. The platform we use is PyTorch. Specifically, we apply ViT and DeiT models implemented in Pytorch Image Models (timm) [7].

---

[7] `https://timm.fast.ai/`

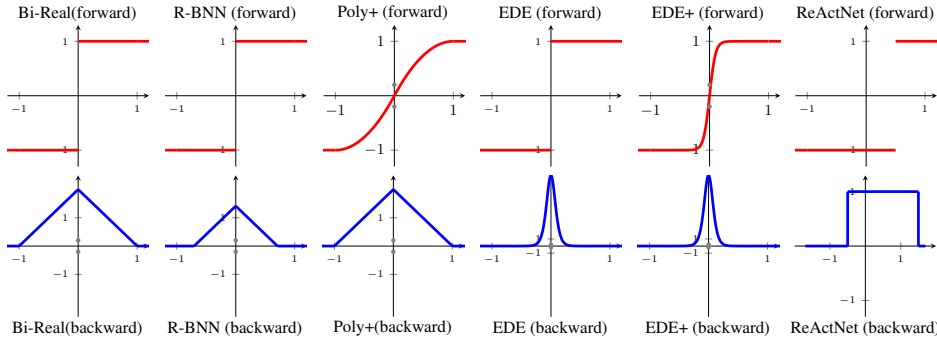

Figure 5: Forward and backward pass for 6 additional ProxConnect++ algorithms.

Table 6: Results for additional ProxConnect++ algorithms on binarizing vision transformers (binarizing weights only), where the results are the mean of five runs with different random seeds.

| Model | Dataset | FP | ProxConnect++ | | | | | | |
|-------|---------|-----|----------|-------|-------|-------|-------|-------|-------|
| | | | Bi-Real | R-BNN | Poly+ | EDE | EDE+ | ReAct | BNN++ |
| DeiT-T | CIFAR-10 | 94.85% | 84.11% | 84.54% | 85.31% | 84.99% | 85.57% | 85.35% | **86.41%** |
| | CIFAR-100 | 72.37% | 59.01% | 59.02% | 60.00% | 59.32% | 60.04% | 60.11% | **60.33%** |
| | **ImageNet-1K** | 72.20% | 64.55% | 64.59% | 64.97% | 64.28% | 65.01% | 65.37% | **67.34%** |
| DeiT-S | CIFAR-10 | 95.09% | 85.01% | 86.07% | 84.99% | 85.37% | 85.33% | 85.91% | **86.19%** |
| | CIFAR-100 | 73.19% | 59.66% | 59.75% | 60.14% | 60.09% | 61.17% | 61.09% | **62.98%** |
| | **ImageNet-1K** | 79.90% | 70.51% | 70.87% | 71.36% | 70.66% | 72.99% | 72.53% | **73.53%** |

**Pre-trained models:**   In this work, we applied pre-trained full precision models for fine-tuning tasks. Here we specify the links to the models we used (note that we choose patch size equal to 16 across all models):

- ViT-B    (ImageNet-1K):    `https://storage.googleapis.com/vit_models/augreg/B_16-i21k-300ep-lr_0.001-aug_medium1-wd_0.1-do_0.0-sd_0.0--imagenet2012-steps_20k-lr_0.01-res_224.npz;`

- ViT-B (ImageNet-21K): `https://storage.googleapis.com/vit_models/augreg/B_16-i21k-300ep-lr_0.001-aug_medium1-wd_0.1-do_0.0-sd_0.0.npz;`

- DeiT-T   (ImageNet-1K):   `https://dl.fbaipublicfiles.com/deit/deit_tiny_patch16_224-a1311bcf.pth;`

- DeiT-S   (ImageNet-1K):   `https://dl.fbaipublicfiles.com/deit/deit_small_patch16_224-cd65a155.pth;`

- DeiT-B   (ImageNet-1K):   `https://dl.fbaipublicfiles.com/deit/deit_base_patch16_224-b5f2ef4d.pth.`

# E    Comparison with BiViT

He et al. [19] propose BiViT, which considers the same binarization task on vision transformers (specifically, Swin-T and Nest-T). He et al. [19] follow a different implementation with softmax-aware binarization and information preservation. To fairly compare with this work, we follow the same setting and run PC++ on Swin-T and NesT-T on ImageNet-1K. We observe that BNN++ achieves 71.3% Top-1 accuracy (BiViT:70.8%) and 69.3% Top-1 accuracy (BiViT:68.7%) respectively on Swin-T and NesT-T. Note that BiViT simply applies BNN as the main algorithm and may be further improved with PC++ algorithms.

# F    Additional forward-backward quantizers

In this section, we summarize additional forward-backward quantizers, which can be either improved or justified with our PC++ framework. Specifically, we find that:

Table 7: Error bar for binarizing weights (BW), binarizing weights and activation (BWA) and binarizing weights, activation, with 8-bit accumulators (BWAA) on CNN backbones. We consider the end-to-end (E2E) pipeline. We compare five variants of ProxConnect++ (BC, PC, BNN, BNN+, and BNN++) with FP, PQ, and rPC. For the end-to-end pipeline, we report the mean of five runs with different random seeds.

| Dataset | Task | FP | PQ | rPC | ProxConnect++ | | | | |
| | | | | | BC | PC | BNN | BNN+ | BNN++ |
|---|---|---|---|---|---|---|---|---|---|
| CIFAR-10 | BW | 92.01%±0.19 | 81.59%±0.11 | 81.82%±0.16 | 87.51%±0.07 | 88.05%±0.05 | 89.92%±0.11 | 89.39%±0.13 | **90.03%**±0.06 |
| | BWA | 92.01%±0.13 | 81.51%±0.16 | 81.60%±0.09 | 86.99%±0.11 | 87.26%±0.23 | 89.15%±0.08 | 89.02%±0.16 | **89.91%**±0.09 |
| ImageNet-1K | BW | 78.87%±0.06 | 63.23%±0.11 | 66.39%±0.22 | 67.45%±0.04 | 67.51%±0.09 | 67.49%±0.12 | 66.99%±0.26 | **68.11%**±0.02 |
| | BWA | 78.87%±0.18 | 61.19%±0.22 | 64.17%±0.19 | 65.42%±0.22 | 65.31%±0.17 | 65.29%±0.21 | 65.98%±0.15 | **66.08%**±0.13 |

Table 8: Error bar on binarizing vision transformers (BW and BWA) on ImageNet-1K. We consider the end-to-end (E2E) pipeline. We compare five variants of ProxConnect++ (BC, PC, BNN, BNN+, and BNN++) with FP, PQ, and rPC. The results are the mean of five runs with different random seeds.

| Model | Task | FP | PQ | rPC | ProxConnect++ | | | | |
| | | | | | BC | PC | BNN | BNN+ | BNN++ |
|---|---|---|---|---|---|---|---|---|---|
| DeiT-T | BW | 72.20%±0.11 | 61.23%±0.07 | 60.35%±0.19 | 63.22%±0.21 | 66.15%±0.11 | 65.00%±0.15 | 66.67%±0.09 | **67.34%**±0.07 |
| | BWA | 72.20%±0.13 | 60.01%±0.12 | 58.77%±0.08 | 62.13%±0.06 | 65.29%±0.19 | 63.75%±0.18 | 65.29%±0.06 | **65.65%**±0.03 |
| DeiT-S | BW | 79.91%±0.21 | 69.87%±0.26 | 68.74%±0.16 | 73.16%±0.19 | 73.51%±0.22 | 73.77%±0.08 | 73.23%±0.11 | **73.53%**±0.13 |
| DeiT-B | BW | 81.81%±0.17 | 72.54%±0.15 | 70.11%±0.23 | 76.55%±0.07 | 76.61%±0.24 | 75.60%±0.17 | 76.63%±0.13 | **76.74%**±0.07 |

- Bi-Real Net [37]/R-BNN [31]: $\mathsf{F}(\mathbf{w}) = \text{sign}$, $\mathsf{B}(\mathbf{w}) = \nabla F(\mathbf{w})$, where $F(\mathbf{w})$ is a piewise polynomial function. We simply choose $\mathsf{F} = F(\mathbf{w})$ and arrive at our legitimate variant Poly+. Note that we gradually increase the coefficient of $F(\mathbf{w})$ such that we ensure full binarization at the end of the training phase.

- EDE in IR-Net [46]: $\mathsf{F}(\mathbf{w}) = \text{sign}$, $\mathsf{B}(\mathbf{w}) = \nabla g(\mathbf{w}) = kt(1 - \tanh^2(t\mathbf{w}))$, where $k$ and $t$ are control variables varying during the training process, such that $g(\mathbf{w}) \approx \text{sign}$ at the end of training. Again, we choose $\mathsf{F} = g(\mathbf{w})$ and arrive at our new legitimate variant EDE+.

- ReActNet [36] can be well justified and is a special case of PC++.

We visualize these forward-backward quantizers and our new variants in Figure 5. Moreover, we perform experiments on vision transformers to examine the performance of additional quantizers and their modified variants. We report the results in Table 6 and observe that (1)Our new proposed Poly+ and EDE+ always outperform the original algorithms and further confirm that our PC++ framework merits theoretical and empirical justifications; (2)BNN++ still outperforms other algorithms on all tasks.

# G  Additional results for end-to-end training

Finally, we provide the error bars for our main experiments in Table 7 and Table 8 for CNN backbones and vision transformer backbones, respectively.

