# OpenReview forum: "Understanding Neural Network Binarization with Forward and Backward Proximal Quantizers"
_NeurIPS.cc/2023/Conference — NeurIPS 2023 poster_

### Official Review · Reviewer_fayS · 2023-06-19

**Soundness:** 3 good
**Presentation:** 3 good
**Contribution:** 3 good
**Rating:** 5
**Confidence:** 3

**Summary:**

The paper generalizes ProxConnect with forward-backward quantizers and introduces ProxConnect++ that includes some binarization techniques as special cases. With the derived ProxConnect++, the paper proposes BNN++ to illustrate the effectiveness of ProxConnect++. Experiments show the advantages of BNN++ on image classification benchmarks with CNNs and vision transformers.

**Strengths:**

The paper generalizes ProxConnect and presents the unified framework ProxConnect++, which includes some binarization techniques as special cases. With ProxConnect++, one may design new forward-backward quantizers. Extensive experiments on CNNs and vision transformers demonstrate the advantages of the proposed ProxConnect++.

**Weaknesses:**

The paper claims that it develops a unified framework ProxConnect++, which allows to design various forward-backward quantizers. However, the forward and backward quantizers used in the paper are limited to the existing ones. To show the advantages of ProxConnect++, it should design new forward and backward quantizers, which outperform the existing ones. The paper only gives an example BNN++, which is also inspired the existing forward and backward quantizers.

**Questions:**

The paper presents a unified framework ProxConnect++. It is claimed that ProxConnect++ can include many binarization techniques as special cases. I have two concerns for the unified framework.

First, it is desirable to present a unified framework. So one can analyze existing algorithms in a unified way. As a unified framework, the most important thing is to propose new superior algorithms following the framework. However, the paper does not give more insights into developing new algorithms. Note that BNN++ is also inspired by the exiting forward and backward quantizers.

Second, the paper compares with some variants of ProxConnect++ and shows that BNN++ performs the best. It is better to include other state-of-the-art binarization techniques in the experiments.


**Limitations:**

The authors addressed the limitations of their work. There is no negative societal impact.

---

> ### Author Rebuttal · Authors · 2023-08-09
>
> We would first like to thank Reviewer fayS for the great review and questions. Below we address your concerns:
>
> **Design new proximal quantizers:**
>
> (1) Prior to our work, existing implementations mostly designed the backward quantizer in an ad hoc fashion (based on graphic approximations of the sign function), while Dockhorn et al. showed that any (continuous) monotonic function can be used as a backward quantizer. Our Theorem 1 allows practitioners to easily design compatible forward and backward quantizers.
>
> (2) We will add the following corollary to facilitate the design of F-B quantizers:
>
> > **If the forward quantizer is a continuous differentiable function (with bounded support), then one can simply choose the backward quantizer as the derivative of the forward quantizer.** This follows from Theorem 1 since $\mathsfit{P}(w) \equiv w$ is clearly proximal. Note that the BNN example does not follow from this Corollary (but still follows from the more general Theorem 1).
>
> (3) Aside from the BNN++ example, we give more examples here on how to derive new F-B quantizers from existing implementations (visualization for existing algorithms and new ones are presented in Figure 1 of the global response):
>
> - Bi-Real/R-BNN: $\mathsf{F}(\mathbf{w})=\text{sign}$, $\mathsf{B}(\mathbf{w})=\nabla F(\mathbf{w})$, where $F(\mathbf{w})$ is a piewise polynomial function. We simply choose $\mathsf{F} =F(\mathbf{w})$ and arrive at our legitimate variant Poly+. Note that we gradually increase the coefficient of $F(\mathbf{w})$ such that we ensure full binarization at the end of the training phase.
> - EDE in IR-Net: $\mathsf{F}(\mathbf{w})=\text{sign}$, $\mathsf{B}(\mathbf{w})=\nabla g(\mathbf{w})=kt(1-\text{tanh}^2(t\mathbf{w}))$, where $k$ and $t$ are control variables varying during the training process, such that $g(\mathbf{w})\approx \text{sign}$ at the end of training. Again, we choose $\mathsf{F} =g(\mathbf{w})$ and arrive at our new legitimate variant EDE+.
>
> Empirically, we find both variants outperform their original form in Table 1 of the global response.
>
> (4) With numerous examples, we have demonstrated that PC++ helps us understand and improve existing gradient approximation methods. We find that using PC++ to design new proximal quantizers based on existing ones is straightforward according to the corollary above. Furthermore, given any other forward quantizers (e.g., higher-order approximation of $\text{sign}$), we can easily derive their corresponding backward quantizers.
>
> **Sota binarization techniques:**
>
> (1) In Appendix E, we compare with another Sota binarization work on vision transformers called BiViT, where we observe BNN++ outperforms BiViT under the same experimental setting.
>
> (2) We add additional results on comparison with other SOTA BNN methods, including Bi-Real Net, Rotated RNN, IR-Net (EDE), and ReActNet. As stated in our previous paragraph, we find that using our PC++ framework, we can improve existing methods with better F-B quantizers design. Furthermore, we find that BNN++ still outperforms other algorithms across all tasks.
>
> (3) In this paper we mainly study the design of forward-backward quantizers.  To verify our theoretical framework on vision transformers, we already applied several empirical tricks to improve the performance of baseline methods and our methods, including the layer-wise scaling factor, additional layernorm function, knowledge distillation, and additional pertaining.
>
> (4) Nevertheless, we are aware of several other techniques such as attention matching, channel re-scaling, shifted activation, state-aware gradient update, and so on, which may boost the performance of PC++ algorithms. We identify this as a limitation of our work and we leave it as future work to further improve our established baselines by combining and adapting some of these techniques.

---

> > ### Author Response · Authors · 2023-08-18
> > **Did our rebuttal address your concerns?**
> >
> > Dear reviewer fayS, as the discussion deadline is approaching, we are wondering if our response has addressed your concerns. We would be happy to answer any of your further questions. Thank you!

---

### Official Review · Reviewer_5jjv · 2023-06-30

**Soundness:** 3 good
**Presentation:** 3 good
**Contribution:** 3 good
**Rating:** 7
**Confidence:** 4

**Summary:**

Thanks to authors for submitting their work to NeuRips 2023. After nicelly flowing introduction the lines 47-58 lay goals of the paper and its contributions out in the context of recent advances, cf. Dockhorn et al. [13], PC (ProxConnect). In particular, the paper generalizes PC to forward-backward binarization, as opposed to PC using only forward binarization, labelling this more general approach PC++. As a practical result of presented theory the fully binarized (8-bit integers) BNN++ is shown to be competitive against existing approaches on most tasks, including experiments on several datasets and architectures (incl. CNN and ViT). It achieves 30x reduction in memory and storage with a modest 5-10% accuracy drop compared to full precision training. Binarization results on vision transformers, rare up until now, is especially timely and promising given the recent deployment of the transformers in deep learning. Architecture ablation experiments conducted demonstrate practicality of the hereby developed theory and increase potential impact of the paper.

Besides Theorem 1, being rather a straightforward extension of previous work Dockhorn et al. [13], including convergence guarantees applied on PC++ in appendix, I value authors used this general result to present a unifying and pragmatic framework for binarization of the NN in response to increasing energy demands for training and fine-tuning the recent large transformer models. I find paper timely and relevant contribution to research community.

**Strengths:**

+ $\textbf{[Accessibility, Approach]}$ The goal of the paper is achieved in very accessible and neat form. Proving Theorem 1 (the main result) introduces a compelling and general framework for designing binarization algorithms for NNs.

+ ProxConnect (3) ++  >>> nice idea of formally adding transformation T that “cancels out” (formally in the Theorem 1 sense) in backward step, so the original theory of PC, Dockhorn et al. [13], applies.

+ $\textbf{[Unifying framework]}$ anAdditional and very impactful merit of the paper is its unifying approach of currently quite scattered and uneasy to access research of neural networks binarization. Especially natural inclusion of the transformers is very timely, convenient and impactful addressed by only few previous works, e.g. Y. He et al. [19].

+ $\textbf{[Demonstration of practical usefulness]}$ (Architecture) Ablation experiments present practical usefulness of hereby developed theory (Theorem 1) (line: 283-284) and increasing the impact of the paper on further research.


**Weaknesses:**

- Is the extension of previous work of Dockhorn et al. [13] novel enough? As noted above in Summary and Strength sections, formally adding transformation $T$ that “cancels out” (formally in the Theorem 1 sense) in backward step, so the original theory of PC, Dockhorn et al. [13], applies is smart yet rather straightforward (also demonstrated in the proof in Appendix). Some may see this rather a direct consequence of the previous work, that should be fully contributed to the respective authors :-)... .Despite this caveat I believe the paper's focus on application of this theory outweighs the theoretical merits and yields a work with potential high impact on further advances in the field. I leave to authors without impacting my decision whether or not to recognise previous work of Dockhorn et al. [13] more and consider renaming the Theorem 1 to "Corollary" to refer directly to previous work.

- Extend of experiments in terms of datasets and modes is well selected. Yet number of runs to avoid statistical error, especially in Tables $3$ and $4$, is insufficient and should be higher. See Questions section.

**Questions:**

- Would it be possible to Increase an extend of randomly seeded pool of experiments for Table 3,4? - reported averages on 3 runs together with rather diminishing differences between columns may not be significant. Because the results are used to draw performance claims, as on line 250, 251, “In particular, for end-to-end training, the best performing ProxConnect++ algorithms achieve ≈ 5% accuracy drop”,  the extended experiments are suggested to be reported in camera ready version. Especially as it concerns ViT, treated rarely before and thus of high impact and interest to community.

- The Figure 2., p.8, presents well why more involved techniques that post-training binarization (PTB) are needed. Would it be worth to use it in the beginning of the paper as a motivation figure? Accompanied with similar figure of memory and storage requirements, perhaps.

- What does Table 2 show? (accuracy?) While it is commented in the text it is not clear from Table captions. I recommend to factor it in for the sake of higher clarity.

- Line 136: “proved” —> “proven”

**Limitations:**

Suggestion: For a camera ready version notes on limitation of the method are suggested to be brought back to main body of the paper from the Appendix G.

---

> ### Author Rebuttal · Authors · 2023-08-09
>
> We would first like to sincerely thank Reviewer5jjv for appreciating our contribution and providing valuable suggestions. Below we address your concerns:
>
> **(1) Novelty on theory:**
> We agree that PC++ is an extension of PC, and we have followed the reviewer's suggestion to label the convergence result in the appendix as a Corollary and credit it more directly to Dockhorn et al.
>
> **(2) More runs to avoid statistical error**:
> We have increased the randomly seeded pool of experiments from 3 to 5 for Table 3 in the main paper. We confirm that the updated Table (denoted as Table 2 in the global response) reports very similar results with 3 runs and our conclusion remains the same. Due to limited time, we can only finish Table 3 in a week but we will update Table 4 as well in our final draft.
>
> **(3) Figure 2**:
> We agree that Figure 2 would serve as a nice motivation for this work. We have modified the original Figure 2 and provided an additional figure on memory footprint (Figure 2 in the global response) to illustrate:
> > BAT is necessary for our context as it performs much better than PTB and reduces the same memory footprint during inference.
>
> (4) Yes, Table 2 shows accuracy. We will add the caption to our final draft.
>
> (5) We have changed "proved" to "proven" on line 136.
>
> (6) Thank you for the suggestion. We will move the limitations back to the main paper in the final draft.

---

> > ### Comment · Reviewer_5jjv · 2023-08-15
> >
> > Thanks to authors for response and adjustments. I believe they will be appreciated by readers. I have no further comments. Thank you.

---

> > > ### Author Response · Authors · 2023-08-18
> > > **Thank you!**
> > >
> > > Dear Reviewer 5jjv, we want to thank you again for your positive review and great suggestions that helped improve our paper. We will make sure that these adjustments are incorporated into our final draft!

---

### Official Review · Reviewer_WcKF · 2023-07-04

**Soundness:** 3 good
**Presentation:** 4 excellent
**Contribution:** 3 good
**Rating:** 7
**Confidence:** 3

**Summary:**

This paper proposes a new framework for training neural networks with binary weights, which generalizes ProxConnect and takes it as a special case. In the new framework, forward and backward quantizers are defined. A consistency result of the two quanziters is derived (Theorem 1). Extensive experiments are conducted to evaluate the usefulness of the new framework.

**Strengths:**

1. The paper is very well-written and easy to the follow. I love the clarity of the paper.
2. The theoretical framework is sensible and the analysis is rigorous. I did not find a technical flaw.
3. The empirical evaluation is sufficient. The code is provided for reproduction.

**Weaknesses:**

1. The proposed framework seems a direct generalization of ProxConnect. Thus, it's novelty is somewhat limited.
2. It is not straightforward for practitioners how to use the framework to design new proximal quantizers. Please provide practical guidelines and tips.

**Questions:**

See above.

**Limitations:**

See above.

---

> ### Author Rebuttal · Authors · 2023-08-09
>
> We first thank Reviewer WcKF for the positive review and great questions, which we address blew:
>
> **Novelty on theory:**
> - We agree with Reviewer WcKF (and also Reviewer 5jjv) that PC++ is an extension of PC, and we are more than happy to credit the theoretical convergence property of PC++ to Dockhorn et al.
> - As pointed out by Reviewer 5jjv, our paper also contributes to designing the unifying framework of forward-backward quantizers, deriving a sufficient and necessary condition, and demonstrating the practical usefulness of our framework.
>
> **Design new proximal quantizers:**
>
> (1) Prior to our work, existing implementations mostly designed the backward quantizer in an ad hoc fashion (based on graphic approximations of the sign function), while Dockhorn et al. showed that any (continuous) monotonic function can be used as a backward quantizer. Our Theorem 1 allows practitioners to easily design compatible forward and backward quantizers.
>
> (2) We will add the following corollary to facilitate the design of F-B quantizers:
>
> > **If the forward quantizer is a continuously differentiable function (with bounded support), then one can simply choose the backward quantizer as the derivative of the forward quantizer.** This follows from Theorem 1 since $\mathsfit{P}(w) \equiv w$ is clearly proximal. Note that the BNN example does not follow from this Corollary (but still follows from the more general Theorem 1).
>
> (3) Aside from the BNN++ example, we give more examples here on how to derive new F-B quantizers from existing implementations (visualization for existing algorithms and new ones are presented in Figure 1 of the global response):
>
> - Bi-Real/R-BNN: $\mathsf{F}(\mathbf{w})=\text{sign}$, $\mathsf{B}(\mathbf{w})=\nabla F(\mathbf{w})$, where $F(\mathbf{w})$ is a piewise polynomial function. We simply choose $\mathsf{F} =F(\mathbf{w})$ and arrive at our legitimate variant Poly+. Note that we gradually increase the coefficient of $F(\mathbf{w})$ such that we ensure full binarization at the end of the training phase.
> - EDE in IR-Net: $\mathsf{F}(\mathbf{w})=\text{sign}$, $\mathsf{B}(\mathbf{w})=\nabla g(\mathbf{w})=kt(1-\text{tanh}^2(t\mathbf{w}))$, where $k$ and $t$ are control variables varying during the training process, such that $g(\mathbf{w})\approx \text{sign}$ at the end of training. Again, we choose $\mathsf{F} =g(\mathbf{w})$ and arrive at our new legitimate variant EDE+.
>
> Empirically, we find both variants outperform their original form in Table 1 of the global response.
>
> (4) With numerous examples, we have demonstrated that PC++ helps us understand and improve existing gradient approximation methods. We find that using PC++ to design new proximal quantizers based on existing ones is straightforward according to the corollary above. Furthermore, given any other forward quantizers (e.g., higher-order approximation of $\text{sign}$), we can easily derive their corresponding backward quantizers.

---

> > ### Author Response · Authors · 2023-08-18
> > **Did our rebuttal address your concerns?**
> >
> > Dear reviewer WcKF, as the discussion deadline is approaching, we are wondering if our response has addressed your concerns. We would be happy to answer any of your further questions. Thank you!

---

### Official Review · Reviewer_YJmg · 2023-07-04

**Soundness:** 3 good
**Presentation:** 3 good
**Contribution:** 3 good
**Rating:** 5
**Confidence:** 4

**Summary:**

This paper studied binary neural networks which extend the existing theory of ProxConnect(PC) to ProxConnect++ and explored the fully binarized scenario, where the dot-product accumulators are also quantized to 8-bit integers. The authors also proposed BNN++ with non-linear forward and backward approximation to the sign function. Authors did experiments on CIFAR10 and ImageNet datasets.

**Strengths:**

Binarizing weights and activations in neural networks is a challenging problem worth studying. This paper proposed an in-depth study of forward and backward functions for binary neural networks and proposed a new method that performs outstanding accuracy on CIFAR10 and ImageNet datasets.

**Weaknesses:**

The authors only compared their method with several BNN methods. But many classic and famous BNN works are not even mentioned, such as XNOR-Net. And some papers studying the backward approximation are missing, e.g., the second-order approximation to the sign function proposed in Bi-Real Net should be compared in Fig.1. I would suggest authors compare or at least mention the binary neural networks to make the paper more comprehensive.

To name a few:

[1] Zihan Xu, Mingbao Lin, Jianzhuang Liu, Jie Chen, Ling Shao, Yue Gao, Yonghong Tian, and Rongrong Ji. Recu: Reviving the dead weights in binary neural networks. (CVPR)

[2] Zhijun Tu, Xinghao Chen, Pengju Ren, and Yunhe Wang. Adabin: Improving binary neural networks with adaptive binary sets. (ECCV)

[3] Zechun Liu, Baoyuan Wu, Wenhan Luo, Xin Yang, Wei Liu, and Kwang-Ting Cheng. Bi-real net: Enhancing the performance of 1-bit cnns with improved representational capability and advanced training algorithm. (ECCV)

[4] Mohammad Rastegari, Vicente Ordonez, Joseph Redmon, and Ali Farhadi. Xnor-net: Imagenet classification using binary convolutional neural networks. (ECCV)

[5] Haotong Qin, Ruihao Gong, Xianglong Liu, Mingzhu Shen, Ziran Wei, Fengwei Yu, and Jingkuan Song. Forward and backward information retention for accurate binary neural networks. (CVPR)

[6] Brais Martinez, Jing Yang, Adrian Bulat, and Georgios Tzimiropoulos. Training binary neural networks with real-to-binary convolutions. (ICLR)

[7] Zechun Liu, Zhiqiang Shen, Marios Savvides, and Kwang-Ting Cheng. Reactnet: Towards precise binary neural network with generalized activation functions. (ECCV)

[8] Chunlei Liu, Peng Chen, Bohan Zhuang, Chunhua Shen, Baochang Zhang, and Wenrui Ding. Sa-bnn: State-aware binary neural network. (AAAI)

[9] Mingbao Lin, Rongrong Ji, Zihan Xu, Baochang Zhang, Yan Wang, Yongjian Wu, Feiyue Huang, and Chia-Wen Lin. Rotated binary neural network. (NeurIPS)

[10] Zechun Liu, Zhiqiang Shen, Shichao Li, Koen Helwegen, Dong Huang, and Kwang-Ting Cheng. How do adam and training strategies help bnns optimization. (ICML)

[11] Hyungjun Kim, Jihoon Park, Changhun Lee, and Jae-Joon Kim. Improving accuracy of binary  neural networks using unbalanced activation distribution. (CVPR)

[12] Koen Helwegen, James Widdicombe, Lukas Geiger, Zechun Liu, Kwang-Ting Cheng, and Roeland Nusselder. Latent weights do not exist: Rethinking binarized neural network optimization. (NeurIPS)


**Questions:**

The authors use the non-linear approximation of the sign function in the forward pass, will that yield other real-valued outputs besides the binary values?

Line 170 is confusing: “BNN++ is more desirable than BNN+ empirically.” The author means BNN++ is better than PC++?

---

> ### Author Rebuttal · Authors · 2023-08-09
>
>
> **Additional References**:
>
> We would first like to thank Reviewer YJmg for providing the additional references, especially Bi-Real net, R-BNN, IR-Net, and ReActNet, which are closely related to our work and expanding our PC++ family. We agree that these are important papers that deserve proper discussion in our work. Here we extend our discussion and we will add them to our paper (we follow the Reviewer's citation order here):
>
> > Existing works propose different methods for improving the performance of BNN and they can be roughly categorized into three classes:
> >
> > (1) different *gradient approximation* that can be either improved or justified with PC++:
> >
> > Some existing implementations [3][5][9] set the forward quantizer as $\text{sign}$ and design the backward quantizer in an ad hoc fashion (based on graphic approximations of the sign function), e.g., [3] applies a piece-wise polynomial approximation; [9] improves [3] with a dynamic polynomial approximation; [5] designs a dynamic error decay estimator based on the $\text{tanh}$ function. These methods in their original forms do not belong to PC++, but they could be brought into the PC++ family by designing new forward quantizers using our Theorem 1:
> >
> > - Bi-Real/R-BNN: $\mathsf{F}(\mathbf{w})=\text{sign}$, $\mathsf{B}(\mathbf{w})=\nabla F(\mathbf{w})$, where $F(\mathbf{w})$ is a piewise polynomial function. We simply choose $\mathsf{F} =F(\mathbf{w})$ and arrive at our legitimate variant **Poly+**. Note that we gradually increase the coefficient of $F(\mathbf{w})$ such that we ensure full binarization at the end of the training phase.
> > - EDE in IR-Net: $\mathsf{F}(\mathbf{w})=\text{sign}$, $\mathsf{B}(\mathbf{w})=\nabla g(\mathbf{w})=kt(1-\text{tanh}^2(t\mathbf{w}))$, where $k$ and $t$ are control variables varying during the training process, such that $g(\mathbf{w})\approx \text{sign}$ at the end of training. Again, we choose $\mathsf{F} =g(\mathbf{w})$ and arrive at our new legitimate variant **EDE+**.
> >
> > Other implementations [2][7] apply shift transformation on both forward and backward quantizers, which belong to the PC++ family.
> >
> > We visualize the forward-backward quantizers of [3][5][7][9], our new Poly+ and EDE+ in Figure 1 of the global response. Moreover, we perform experiments on vision transformers to examine the performance of these 6 additional quantizers in Table 1, we observe that:
> > - Our new proposed Poly+ and EDE+ always outperform the original algorithms and further confirm that our PC++ framework merits theoretical and empirical justifications.
> >- BNN++ still outperforms other algorithms on all tasks.
> >
>  >(2) architecture design that can be further integrated into our PC++ as future work: [1] designs a rectified clamp unit to address "dead weights"; [4] applies the absolute mean of weights and activations; [6] uses real-to-binary attention matching and data-driven channel re-scaling; [11] proposes a shifted activation function.
> >
> > (3) optimization refinement, which again could be integrated into our framework as future work: [8] proposes to rescale the backward gradient on binary activations in order to stabilize BNN training; [10] provides a weight decay scheme; and [12] proposes Bop, a new optimizer for BNNs.
> >
> **Questions**:
>
> (1) **Full binarization:** In our experiments (lines 195-196), we linearly increase $\mu$ in BNN++ to achieve full binarization in the end. To avoid real-valued outputs, we also performed a sanity check to confirm the final weights are 100% binary.
>
> (2) **Regarding the relationship between BNN++ and PC++**:
>
> Here we clarify that BNN++ is an algorithm that belongs to the broader PC++ family:
> - Firstly, we want to emphasize that PC++ is a family of algorithms where the forward/backward quantizers satisfy the condition in Theorem 1;
> - Secondly, as we discussed in Example 2, BNN+ cannot be justified under the framework of PC++, thus does not belong to PC++.
> - Thirdly, our new variant BNN++ is designed to be a legitimate PC++ algorithm and is empirically shown to perform better than BNN+.
> - Similarly to BNN+, references [3][5][9] provided by the Reviewer use different backward quantizers and the same $\text{sign}$ function, thus cannot be justified by PC++. Using our Theorem 1, we design new algorithms Poly+ and EDE+ that belong to the PC++ family. These new variants are able to explore the loss landscape with gradients evaluated at more fine-grained weights, especially in the initial phase of training (as opposed to evaluating the gradient at full quantized weights).

---

> > ### Author Response · Authors · 2023-08-18
> > **Did our rebuttal address your concerns?**
> >
> > Dear reviewer YJmg, as the discussion deadline is approaching, we are wondering if our response has addressed your concerns. We would be happy to answer any of your further questions. Thank you!

---

### Author Rebuttal · Authors · 2023-08-09

We would like to thank all reviewers again for their extremely informative reviews that helped us improve the paper. Here we want to provide additional figures and tables (in the new one-page PDF file) that we will add to our final draft:

(1) **Figure 1:** According to Reviewer YJmg's suggestions, we have carefully reviewed more existing methods and summarized additional forward-backward quantizers here, which can be either *improved* or *justified* with our PC++ framework. Specifically, we find that:
- Bi-Real Net, R-BNN, and Error Decay Estimator (EDE) in their original forms do not belong to PC++, but they can be improved by modifying the forward quantizers (we will show the improvement in the next table). Thus, we propose the modified Poly+ and EDE+.
- ReActNet is a special case of PC++.

(2) **Table 1:** Following Figure 1, we perform experiments on vision transformers to examine the performance of additional quantizers (and their modified variants), we observe that:
- Our new proposed Poly+ and EDE+ always outperform the original algorithms and further confirm that our PC++ framework merits theoretical and empirical justifications.
- BNN++ still outperforms other algorithms on all tasks.

(3) **Figure 2:** Following the suggestions by Reviewer 5jjv, we modify the original Figure 2 and provide an additional figure on memory footprint to serve as a motivation for this paper:
> BAT is necessary for our context as it performs much better than PTB and reduces the same memory footprint during inference.

(4) **Table 2:** As suggested by Reviewer 5jjv, we increase the randomly seeded pool of experiments from 3 to 5 for Table 3 in the main paper. We confirm that the updated Table reports very similar results with 3 runs and our conclusion remains the same.

---

### Decision · Program_Chairs · 2023-09-21

**Decision:**

Accept (poster)

**Comment:**

All reviewers lean towards acceptance. The paper puts forward a principled method to design forward-and-backward quantizers for binarized networks. Besides good theorical contributions, empirical results show noticeable and consistent improvements. The authors are encouraged to incoporate missing citations suggested by the reviewers, and potentially add them for experimental comparisons if appropriate.